# Beyond the pill: Understanding barriers and enablers to oral and long-acting injectable PrEP among women in sex work in Zambia

Ramya Kumar[1,2]*, Chisomo Mwale[3], Patricia Maritim[4], Jamia Phiri[2], Wendy Barrington[1,5,6], Ruth Zyambo[7], Martin Zimba[8], Kenneth Mugwanya[1,9], Michael Herce[2,10], Maurice Musheke[2], Deepa Rao[9‡], Anjali Sharma[2‡]

1 Department of Epidemiology, School of Public Health, University of Washington, Seattle, Washington, United States of America, 2 Centre for Infectious Disease Research in Zambia (CIDRZ), Lusaka, Zambia, 3 Department of Development Practice, Laney Graduate School, Emory University, Atlanta, Georgia, United States of America, 4 University of Zambia, School of Public Health, Lusaka, Zambia, 5 Child, Family, and Population Health Nursing, School of Public Health; University of Washington, Seattle, Washington, United States of America, 6 Health Systems and Population Health; School of Public Health; University of Washington, Seattle, Washington, United States of America, 7 Tithandizeni Umoyo Network, Lusaka, Zambia, 8 Zambia Sex Workers Alliance, Lusaka, Zambia, 9 Department of Global Health, School of Public Health, University of Washington, Seattle, Washington, United States of America, 10 Institute for Global Health and Infectious Diseases, University of North Carolina, Chapel Hill, North Carolina, United States of America

‡ Co-senior authors.
* Ramya.Kumar.MLK@gmail.com

## Abstract

Women engaging in sex work (WESW) in low- and middle-income countries face a disproportionately high risk of HIV infection. This study explores enablers and barriers to the uptake and persistence of oral pre-exposure prophylaxis (PrEP) and long-acting injectable PrEP (LAI-PrEP) among WESW in Lusaka, Zambia. We evaluated Capability, Opportunity, and Motivation behavioral domains, using the COM-B model, which affected behavioral engagement with PrEP services among newly-initiated WESW from community-based safe spaces. Participants were recruited from July—October 2023 and interviewed using a semi-structured guide to explore barriers and enablers to engagement with HIV prevention. We used a rapid analysis approach—a two-step qualitative method—to identify themes aligned with COM-B domains. Interviews were conducted in English, ChiNyanja, or IchiBemba, audio-recorded, translated into English when necessary, and transcribed verbatim. Among 18 participants with a median age of 28 years (IQR:23–33) and 5 years in sex work (IQR:2,7), education during outreach by peer navigators and program staff was crucial to building trust and demystifying PrEP. Persistent knowledge gaps and misconceptions, especially about daily adherence and alcohol use, were significant barriers. Trustworthy program staff and reliable service provision facilitated continued PrEP use, and women preferring that drugs be delivered to them. Social support systems were mixed, offering both aid and competition. Personal empowerment and health protection motivated PrEP use, with LAI-PrEP preferred for eliminating daily pill burdens and associated stigma. However, inconsistent

**Data availability statement:** Due to the sensitive nature of this research involving a small sample of marginalized women and the importance of protecting participant confidentiality, the interview data are not publicly available. De-identified transcripts or other supporting materials may be made available to qualified researchers upon reasonable request, contingent on ethical approval and in accordance with institutional review board guidelines. Requests should be directed to the corresponding author or to the Regulatory Department at the Centre for Infectious Disease Research in Zambia (CIDRZ) via email at: regulatory@cidrz.org.

**Funding:** The work was supported by an operating grant from the National Institutes of Health Fogarty Global Health Fellowship awarded by the National Institutes of Health Fogarty International Center Grant (#D43TW009340 to RK). The funders had no role in study design, data collection and analysis, decision to publish, or preparation of the manuscript.

**Competing interests:** The authors have declared that no competing interests exist.

supply and misconceptions about LAI-PrEP were potential barriers. This study underscores the importance of person-centered care in addressing the complex interplay of individual, community, and programmatic factors influencing PrEP engagement among WESW in Zambia. A holistic focus, and adaptive health service delivery approach are both crucial to ensure that advances in HIV prevention translate into tangible benefits for WESW. Reliable, respectful healthcare programs that provide accurate, and trusted information are essential for improving PrEP uptake and persistence.

## Author summary

### What is already known on this topic

Women engaging in sex work (WESW) in low- and middle-income countries face a high risk of HIV acquisition, and PrEP is recognized as an effective preventive measure. While willingness to use PrEP among WESW has been documented in clinical trials, barriers to real-world uptake and persistence—including stigma, discrimination, and knowledge gaps—remain largely unexplored, especially regarding the newer long-acting injectable PrEP (LAI-PrEP).

### What this study adds

This study identifies key enablers and barriers to PrEP uptake and persistence among WESW in Zambia, in advance of being the first African country to offer cabotegravir in February 2024. It reveals that education by trusted peer navigators, reliable service provision, and the desire for personal empowerment facilitates PrEP use, while persistent knowledge gaps, misconceptions, and inconsistent supply hinder it. The study also finds a strong preference for LAI-PrEP due to its convenience and reduced stigma, but flags concerns about potential barriers like supply chain inconsistency and misconceptions.

### How this study might affect research, practice, or policy

These findings highlight the need for holistic, community-based interventions tailored to the unique needs of WESW. Addressing misconceptions, improving service reliability, and incorporating peer-led education can enhance PrEP uptake and persistence. Policymakers and health programs should consider these factors when developing future HIV prevention strategies, ensuring that the rollout of LAI-PrEP is both effective, and contextually appropriate.

## Introduction

The global HIV burden among women engaged in sex work (WESW) is alarmingly high, particularly in the Eastern and Southern African regions. In 2022, HIV prevalence was four times as high among sex workers compared with adults in the general

population (aged 15–49 years) [1]. Globally, WESW have a HIV prevalence of 10.4%, and among those living in Eastern and Southern Africa, the HIV prevalence is three times higher at 33.3% [2]. Failure to protect this key population (KP) of WESW from HIV acquisition will prolong the pandemic at a huge cost to the affected communities and broader societies [2].

Pre-Exposure Prophylaxis (PrEP) is a highly effective biomedical intervention recommended by the World Health Organization for anyone at substantial risk of HIV infection. The willingness of WESW to use PrEP has been demonstrated in multiple clinical trials, but this willingness does not always translate into actual use in "real world" settings. Several studies show that uptake of PrEP among sex workers is high, but the persistence on PrEP is low [3–5].

Some barriers to uptake and persistence are deeply entrenched in social perceptions and structural barriers. These barriers include socio-structural factors like stigma, discrimination, criminalization, violence, and financial insecurity [5–8]. Stigma, related to PrEP and being perceived as HIV-positive, affects uptake and persistence on the drug, even during periods when a person is still at risk. There is additional misinformation about PrEP's safety, side effects, and efficacy. Logistical and practical barriers to oral PrEP use also remain. These include the need to frequently test for HIV every 3 months while on PrEP, and individual-level challenges (i.e. unpredictable schedules for sex work, or having to travel out of town to meet clients) which make clinic visits and consistent PrEP usage difficult.

The introduction of long-acting injectable PrEP (LAI-PrEP) throughout the African continent signifies a potential shift in HIV prevention strategies, though its impact is yet to be fully understood. LAI-PrEP has shown superior efficacy over daily oral PrEP in clinical trials, offering a new avenue for prevention [9,10]. The HPTN 084 trial, conducted in seven African countries among HIV-uninfected cisgender women, and the HPTN 083 trial, which enrolled cisgender men who have sex with men, and transgender women who have sex with men across multiple global sites, both showed strong protective effects [9,10]. LAI-PrEP is initially administered once a month for two months, and then once every two months, making its discrete, extended-duration formulation especially appealing to adolescents and young women [11–14]. Nevertheless, the most appropriate ways to implement it remains uncertain given the paucity of information on the perceptions and preferences of WESW with regards to LAI-PrEP. Healthcare services that neglect the specific preferences of WESW within broader HIV prevention programming for KP could hinder the effectiveness of LAI-PrEP.

Zambia's early adoption of LAI-PrEP marks a significant step in prioritizing advanced HIV prevention methods. Zambia was one of the first countries on the African continent to roll out LAI-PrEP in February 2024, and although the roll out prioritized young women and adolescents in the general population, many may have intersecting identities that make them KP [15]. Programs like the PEPFAR-funded Key Population Investment Fund (KPIF) have been successfully engaging with KPs in Lusaka Province, and providing community-based HIV prevention & treatment services to them. KPIF is implemented by the Centre for Infectious Disease Research in Zambia (CIDRZ) in partnership with the Zambian Ministry of Health (MoH), U.S. Centers for Disease Control and Prevention (CDC) and most importantly, Key Population Civil Society Organizations (KP-CSOs). Although programs like KPIF are increasingly reaching sex workers, barriers to PrEP uptake and persistence remain. To address the multilevel barriers that persist in PrEP uptake, health systems must seek to understand these challenges in order to maximize reachable moments— critical points in time when an individual is most open to receiving health interventions, support, or behavior change messaging. Recognizing and leveraging these moments when individuals are most receptive, is an opportunity to provide highly effective prevention methods.

Our study aimed to understand the perceptions and preferences of oral and LAI-PrEP among cis- and transgender women engaged in sex work, as well as peer navigators who identify as WESW and were employed by the KP-CSOs. Specifically, we qualitatively explored the enablers and psychosocial barriers to initiating and persisting on PrEP, and assessed the interest and acceptability of LAI-PrEP. This study is part of a larger parent study that also includes quantitative data to measure the association between multiple stigmas and oral PrEP initiation and persistence among HIV-negative adult WESW [16]. The qualitative findings of this paper will complement and contextualize findings from the quantitative results [17,18].

The findings from this research are crucial for refining HIV service delivery approaches and ensuring the effectiveness of PrEP interventions, ultimately shaping future HIV prevention policies and community-based programs.

## Methodology

### Ethics statement

Written informed consent in English, or one of two local languages: ChiNyanja or IchiBemba, was obtained before enrolment. As added protection for this marginalized population, participants completed an informed consent quiz to ensure that they understood the potential risks to study participants. Each participant received the Zambian Kwacha equivalent of $5.50 (US$1 USD ≈ ZMW18) as compensation for their time and willingness to participate.

This study was approved by the University of Zambia Biomedical Research Ethics Committee (protocol 3650–2023), dated 21/06/2023 and the University of North Carolina (UNC) Biomedical Institutional Review Board (protocol 22–3147), dated 11/07/2023. The Zambian National Health Research Authority, Lusaka Provincial, and Lusaka District Ministry of Health provided written approval for this study. Additionally, the Lusaka District key populations specialist, civil society leaders, and sex work community members oversaw the study through a community advisory board. Details on the board and the co-creation of the study protocol, informed consent, and survey instruments are outlined in the protocol [16]. Additional information regarding the ethical, cultural, and scientific considerations specific to inclusivity in global research is included in the Supporting Information [S2 Checklist].

### Study design

The WiSSPr parent study protocol is published separately, but briefly, this was a mixed-methods study that used a convergent parallel design to collect quantitative and qualitative data concurrently and separately, prioritizing both the strands equally, but keeping them independent during analysis [16]. This paper reports only on the qualitative data.

### Theoretical frameworks

We used the COM-B model to analyze Capability, Opportunity, and Motivation as drivers of behavioral engagement with PrEP services [19–21] [Fig 1]. This framework allowed us to align intervention strategies directly with identified deficits in knowledge or skills (Capability), environmental and social contexts (Opportunity), and personal motivations and attitudes (Motivation). 'Individual' factors are predominantly linked to Capability and Motivation, but can overlap with Opportunity when considering the individual's access to resources. 'Environmental' factors are mostly categorized under Opportunity, reflecting how external conditions facilitate or hinder behavior. 'Programmatic' factors influence both Opportunity and Capability by altering the environment, or by providing education and support that enable the behavior.

The COM-B model, utilized in HIV prevention, proposes a structured cascade approach to identify and address gaps in motivation, access, and effective use of HIV prevention methods within prioritized populations. This model facilitates routine monitoring, advocacy, and the establishment of clear targets akin to the 95-95-95 goals set for HIV treatment. By analyzing reasons for discrepancies at each stage of the prevention cascade—motivation, access, effective use—it offers a framework to support the development and implementation of targeted interventions, thereby potentially enhancing the effectiveness and reach of HIV prevention programs [22].

In practice, identifying factors at various levels (individual, environment, and programmatic) nested within domains of the COM-B model allows for a comprehensive analysis that addresses the specific barriers and facilitators to PrEP uptake and adherence. This integrated approach ensures that all relevant aspects of behavior change are considered, leading to more effective and sustainable health outcomes.

### Key Population Investment Fund background

The WiSSPr parent study was nested within the KPIF program. In the KPIF program, KP-CSOs employ women from the sex work community, known as 'peer-navigators'. Peer navigators leverage their own social networks to recruit sex workers to access health services at community-based wellness spaces, also referred to as 'drop-in centers'. These drop-in

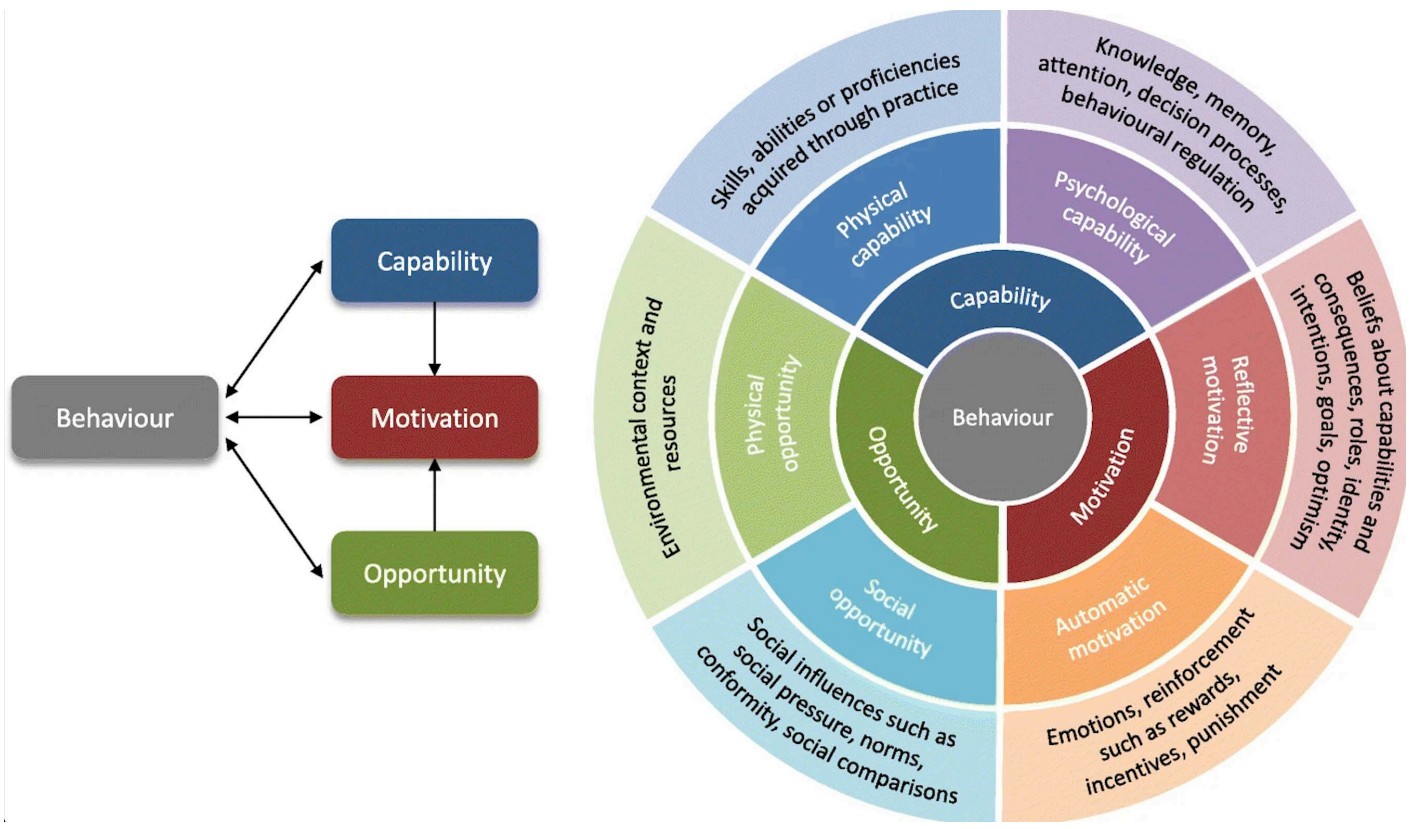

**Fig 1. The COM-B model. This figure is reprinted from McDonagh et al. [21].**

centers have been functioning as community-based outposts of neighboring MoH clinics since October 2021. KPIF also provides venue-based outreach to deliver HIV testing services and prevention services like oral PrEP, condoms and lube at venues where WESW socialize, such as brothels, bars, or the home of a Key Informant (KI). The KI is often a 'queen mother', a leader of WESW, or another trusted woman in the sex work community.

Peer-navigators are involved in all aspects of PrEP delivery, together with MoH clinical and laboratory staff, all of whom are trained to be KP-friendly. Together, the teams liaise with the key informants (KIs) who are respected women in the sex work community, providing health education, HIV testing, PrEP refills, and PrEP adherence support. Their support builds trust and belief in the benefits and effectiveness of PrEP.

Women are given the date for their next PrEP refill, and then connected to a peer navigator who is supposed to call and remind them to pick up their refill either at the drop-in center, or when the program is scheduled to visit their preferred venues.

## Participant selection

From 19 July – 18 October 2023, the parent study recruited WESW who were newly initiated on PrEP from two drop-in centres (i.e. community-based wellness spaces) in Lusaka, Zambia.

We purposively sampled 18 women: 6 who discontinued PrEP immediately after initiation, 6 who continued on PrEP, and 6 peer navigators who provided the perspectives of women who are 'boundary spanners'—at the unique interface of supporters of health service delivery, as well as recipients of care as sex workers themselves. No participants refused to

be interviewed. To determine whether thematic saturation had been reached, we reviewed interview memos after each interview and extracted themes into a matrix on MS Excel (Redmond, WA, USA).

## Data collection

We used a semi-structured interview guide [S1 Text] with open-ended questions and probes to explore specific themes related to HIV prevention and intersectional stigma. This guide allowed interviewers the flexibility to follow topics which were of interest to the participants. The themes that we explored were informed by the Socioecological model and the domains of the COM-B framework, which includes perceived and enacted stigma and discrimination, the impact of intersectional stigmas on health service utilization, service needs, enablers such as psychosocial support or resilience, and building trust in the healthcare system. The guide also included patient education on oral PrEP vs LAI-PrEP. The interviewer explained that LAI-PrEP is a highly effective HIV prevention method for at-risk adults and adolescents, that is administered as a bi-monthly injection to maintain protective medication levels in the blood, similar to oral PrEP, and that LAI-PrEP will be available in Zambia in 2024. We did this to assess participants' perceptions on the advantages and disadvantages of these different PrEP options, as well as their willingness to use them. Participants were asked about their own perceptions, as well as those of other sex workers in their community, because this approach has yielded rich responses in other studies with sex workers [23].

All interviews were conducted by a single trained interviewer. JP is a Black cis-gendered Zambian woman in her mid 20s, with a diploma in social work and social welfare. She was not a sex-worker, and was not from the same socioeconomic background or community as the interviewees. She had 3 years of qualitative interview experience as a research assistant, and had previously interacted with the participants through the wellness spaces where she worked with peer-navigators on programmatic activities. She also administered a quantitative survey to the women at their baseline visit in the study. Participants were verbally consented to the study, and were informed that these 1-hour interviews were to understand what the challenges were for WESW to initiate PrEP, and challenges to continue using it, in order to help the program develop appropriate service delivery strategies. The interview guides were developed with input from a community advisory board and piloted with board members, as well as senior peer-navigators. The interviews were conducted with caution in a private room at the community wellness drop-in centre, or in the private homes of senior members of the sex work community. The well-being of WESW was the central consideration of the study, a driver of the study design, and the reason for the involvement of community advisory board. CSO staff and key informants were not present in the interview room, but they did monitor the emotional reactions of WESW after the interview and provided psychological support to the participants if needed. No repeat interviews were done.

Our positionality as researchers, including our socio-cultural backgrounds, education, and professional roles, influenced both the design of the interview questions and our interpretation of the results. We acknowledge that these factors may have shaped the framing of our questions, as well as our understanding of participants' responses, and we sought to address this by incorporating feedback from a community advisory board and peer navigators throughout the research process.

Interviews were conducted in English, ChiNyanja, or IchiBemba based on participant preference. We requested permission to audio record interviews for transcription and translation purposes. The participants had the right to accept or refuse audio-recording, and all participants opted to be audio-recorded. Audio-recordings were translated into English and transcribed verbatim in a single step by qualified research staff. The audio-recordings were not marked with any identifying information. Instead, a unique qualitative study identifier was used to label the recordings.

## Rapid qualitative data analysis

Given the initial rollout of LAI-PrEP in Zambia in February 2024, and the urgency of releasing these findings to inform implementation, our analysis was conducted in a two-step process using a rapid qualitative analytic approach [24–26]. We used an Interpretative Phenomenological Analysis approach, which focuses not only on describing participants'

experiences, but also on interpreting the underlying meaning within their specific context [27]. Rather than presenting participant statements in isolation, we sought to understand the latent meaning – the deeper influences shaping their perspectives. First, we used a standard template to create a summary memo of each interview, incorporating both the audio recording and the interviewer's field notes. Second, we created an analysis matrix for the summary memos; each row was a unique question, and each column was the individual interview.

We created a codebook with *a priori* high-level categories which were determined based on the Socioecological model nested within the domains of COM-B. We identified themes and subthemes across all participants for each question. After the initial round of deductive theme identification, we mapped the themes onto the appropriate domains of the COM-B model [S1 Data]. Team members met to discuss and reach a consensus on pertinent themes. Members who did not participate in the interview participated in the analysis, and reviewed the findings to confirm that the themes reflected their experiences.

We present the themes, sub-themes, and number of interviews in which themes occurred within a table. We note when a theme was a barrier (−), or enabler (+) to using PrEP, or even both (−/ +), depending on the circumstance. The credibility and trustworthiness of qualitative data were assured through member checking by the participants themselves [28].

### Critical reflexivity statement

Our research team, comprised of public health experts, epidemiologists, and qualitative researchers, acknowledges the potential biases stemming from our external position to the female sex worker community in Lusaka [S1 Checklist]. Recognizing the power dynamics involved, we engaged KIs in the sex work community, who also routinely work with program staff, to foster trust and enhance data authenticity. As academic orientations may shape data interpretation, we involved a community advisory board to inform our analysis processes [16]. This board included both sex workers and healthcare providers, ensuring that findings resonate with the participants' lived experiences. The socio-legal context of sex work in Zambia likely influenced participant responses, given the stigmatization and criminalization of sex work, as well as stigma around HIV prevention methods like PrEP. We continually reflected on these dynamics, discussing the influence of our positionalities, and the research context on our interactions and findings. Through these efforts, we aimed to conduct our research ethically and effectively, fully aware of the influence our backgrounds and the study environment have on the research outputs.

## Results

### Overall summary

Participants had a median age of 28 years [IQR:23,33] with a median of 5 years in sex work [IQR:2,7]. A complex interplay of factors at individual, community, and programmatic levels influences PrEP engagement among WESW which are summarized in Table 1 [Table 1].

### Determinants of oral PrEP

**Capability. Theme: Education about PrEP from peer-navigators and program staff during outreach visits in the community give WESW the knowledge to initiate on PrEP.** The KPIF program's outreach to venues where sex workers socialize was crucial in educating WESW about HIV protection. Peer promoters and staff facilitated these initiatives, demystifyied PrEP, and built trust in its efficacy.The effectiveness of outreach visits is illustrated by a participant who decided to take PrEP after understanding its protective benefits:

> "After I was tested [for HIV], and the result was negative, they told me how PrEP works, so I thought to myself, 'it's good because it protects against HIV.'" (4004, 19 years, continuer)

Table 1. Themes Determinants of PrEP Engagement among Sex Workers. *Key*: Barriers (−), enablers (+), both (− / +); [occurrences].

| Domain | Health System Level | Oral PrEP | LAI-PrEP |
|---|---|---|---|
| Capability | Provider | Education during outreach to venues where sex workers socialize (+) [7] | Misconceptions and knowledge gaps among peers and peer-navigators (−) [3] |
| | Individual | Knowledge gaps (−) [3] | |
| | | Alcohol use incompatible with daily pill adherence (−) [6] | |
| Opportunity | Health system | Trust-worthy and respectful program staff (− / +) [9] | Inconsistent supply chain availability (−) [2] |
| | | Convenience and reliability of service provision (+) [10] | |
| | Individual | Burdens of opportunity costs, financial strain, and side effects (−) [6] | |
| | Community | Sex worker social support system (− / +) [3] | Sex worker social support system (− / +) [3] |
| Motivation | Individual | Observability of effectiveness (+) [3] | Elimination of pill burden (+) [18] |
| | | Protection in work environment of known HIV acquisition risk (+) [13] | |
| | | Compatible with lifestyle (+) [3] | Compatible with lifestyle (− / +) [8] |
| | Community | PrEP Stigma (−) [7] | PrEP Stigma (+) [7] |

The trust and reassurance from peer promoters and program staff was pivotal for many WESW. One participant reflected on the program stating,

> *"The words I was told, I knew [program staff] wouldn't lie to me, and I knew PrEP was useful." (6005, 26 years, continuer)*

The program's strength lied in reaching people where they were (e.g., bars and brothels), but building trust takes time, and not all women started the drug after the first interaction:

> *"They found us in a bar, then talked to us about PrEP, but I didn't start right away because we thought they were lying to us ..." (6007, 36 years, discontinued)*

Despite this, the education provided in these outreach visits was often the turning point for WESW, as noted by one participant: *"They taught us about PrEP when they came to my area. After being educated on it, that's when we knew what it's for. That's when we started taking PrEP." (4005, 39 years, discontinued)*

**Theme: Knowledge gaps about oral PrEP.** Some participants misunderstood the purpose and duration of PrEP versus antiretroviral therapy (ART), revealing knowledge gaps that could hinder effective PrEP use. One woman's story illustrated a common misconception that PrEP, like ART, must be taken indefinitely, which can discourage uptake and persistence.

> *"I just [took oral PrEP] for 2 weeks. I started having stomach cramps, so my friend started telling her friends that I got medicine, and it was prevention for HIV. She [friend of a friend] said ARVs (antiretrovirals) and PrEP are the same because you take it forever until you die. If you stop, then the HIV virus enters. So I became scared and I stopped." (6008, 27 years, discontinued)*

This further illustrated how stigmatization of HIV-related medication complicates one's decision to continue PrEP. Embarrassment and fear of being judged can lead to discontinued PrEP use. The persistence of misinformation and stigma

showed that while outreach programs were effective, continuous education and support were essential to address misconceptions and encourage sustained PrEP adherence.

Misconceptions regarding correct PrEP usage, including beliefs about its lack of effectiveness if it was not taken at the same time daily, or if it was taken with alcohol, also persisted:

*"[oral PrEP] wants [to be taken] every day at the same time." (6002, 35 years, peer-navigator)*

Another highlighted concerns related to alcohol consumption

*"also taking the pills with beer is not good, so it's better in the body." (6005, 26 years, continuer)*

These beliefs may have stemmed from their experiences with, or knowledge of ARV use – despite PrEP not needing to be taken at the same time every day, and it's effectiveness, even when consumed with alcohol.

**Theme: Alcohol use is part of the lifestyle, but incompatible with daily pill adherence.** For many sex workers, alcohol was part of the work culture. It was a job requirement for attracting clients at bars, nightclubs, and other venues. These venues served alcohol and in order to fit in, women buy drinks in order to feel relaxed, meeting clients. Clients then continued to buy drinks, both for themselves, and the sex worker. There was a partnership between the owners of these venues and sex workers; encourage clients to spend on alcohol and increase profits for the venue owners.

In this way, alcohol use was a fundamental part of the lives of sex workers. When interviews took place during the day, some women were hungover and smelled like alcohol they drank the previous night. Alcohol consumption was also a coping mechanism for the violence, rape, and discrimination that WESW faced. One sex worker who discontinued PrEP described how alcohol brought her joy:

*"What brings me joy, even when am going through issues... is alcohol!!! After I take it, I feel very happy, and then I go home to laugh some with my children and then it makes me [more] happy." (6009, 22 years, discontinued)*

One women who immediately discontinued PrEP after initiating, illustrated this point:

*Interviewer: "When you look at all the challenges that you have mentioned where people call you "a whore", and men refusing to pay you … What helps you to have less stress from the challenges you have mentioned?"*

*Participant: "I just drink mooba (term for any drink that contains alcohol)!" (4009, 23 years, discontinued)*

Among women who were consuming substantial amounts of alcohol, their capability to take the drug was compromised, because alcohol impaired cognitive functions like memory, and affected their ability to remember to take their medication as prescribed. One peer-navigator described her relationship with alcohol and PrEP adherence:

*"Forgetting when I'm drunk. When I forget, I miss doses, and PrEP is not supposed to be taken like that. It wants [to be taken] every day at the same time." (6002, 35 years, peer-navigator)*

**Opportunity.  Theme: Trust-worthy, and respectful program staff.** Medical mistrust, and the lack of trust in healthcare systems and providers, was a significant barrier to accessing and utilizing health services. Negative interactions with healthcare providers, including perceived or actual discrimination, bias, and inadequate communication, reinforced this mistrust. Stories of mistreatment spread within the communities, and compounded the sense of distrust.

Many women reported profound discrimination where nurses and doctors verbally abused them for being sex workers, and chased them away from the clinics where they seek care. Women highlighted demotivating experiences when attempting to access healthcare services from government clinics compared to the KPIF program's wellness centres. As one participant described:

*"[The program] care[s] for us. [unlike] when we go to other [government] clinics, they say things like 'Get out of here, we are not the ones who taught you prostitution!' We know we will be given what we want when we go there [to the hub]. They treat us like people." (6004, 31 years, continuer).*

WESW who experienced high levels of discrimination from health care providers were less likely to seek preventive care, adhere to prevention options, or engage with healthcare providers. One peer-navigator recounted how discrimination at public health care facilities impacted the accessibility and reliability of the health services.

*"I once went to a certain government facility. I didn't hide that I was a [sex worker]. Then [the nurse] laughed… went out and brought her friend. She told her colleague that I was a whore. From then on, I stopped {going there}." (6002, 35 years, peer-navigator)*

Women were clear about the need for more KP-friendly spaces, which played a significant role in their decision to initiate and persist on oral PrEP. As one participant reflected:

*"… the [government] clinics, they are not good. It would be better if they built a clinic [specifically] for us. That would be much better because you [ program] people understand us. They don't." (6009, 22 years, discontinued)*

Empathy, respect and effective communication played a critical role in making health services trustworthy. Women commended the program for their respectful, patient, and clear manner of teaching about PrEP, which not only facilitated learning, but also fostered a trusting environment. The rapport that women built with program staff during outreach visits, motivated them to continue using services at the drop-in centers.

*"I chose this space because of how you [program staff] take me. You give me respect, so I feel good, you teach us calmly about PrEP in a proper manner." (4007, 31 years)*

**Theme: Convenience and reliability of service provision**. Women had a strong preference for accessing PrEP from KI homes, and venues where KP socialized (i.e.,: brothels, bars), rather than from government health facilities. Knowing that the KPIF program was reliable and accessible when needed, was a facilitator to PrEP continuation. Although program staff encouraged women to travel to the drop-in center to pick up their refills, staff from the KPIF hub would also conduct outreach visits to offer HIV testing and deliver PrEP refills, which relieved womens' mental anxiety about having to make the time, and find the money to travel to a clinic. Taken together, the convenience of a drop-in center close to a home, and the reliable PrEP delivery (if requested), was an important facilitator for PrEP use:

*"It's near yes, it's available, and as for me, I don't panic because I know that they will bring for me. Even if I have a shortage, they will bring for me." (6006, 21 years, continuer)*

This reliability of services to WESW is due, in part, to the rapport the peer-navigators had with the government clinic staff who were specially trained to provide KP-friendly services in partnership with the KPIF program. One peer-navigator

described the confidence that she had that the head nurse in charge of the partner government facility would not only be receptive to her requests, but organize the delivery of condoms to the community:

> "*There are times when you create a rapport, whereby, you just call [on the phone], 'Sister I want you to bring me condoms.' They bring for you.*" (4002, 26 years, peer-navigator)

Although HIV testing and oral PrEP services are perceived as reliable and accessible, gaps still remain. A significant barrier to discontinuing PrEP was the lack of clear communication regarding the next refill dates, and the absence of reminders. The program was supposed to provide women with a written card with the date when their next refill is due, and women were told to come to the drop-in center if they run out of drug and an outreach visit was not scheduled before that date. One woman described her confusion on whether the program could deliver to her, because she did not know the date of refill, or how to get refills from other places like private chemists (i.e pharmacies).

> "*Where do we get [PrEP] if they don't bring it for us? We always hope that you will bring for us, but if you become quiet, then that means we will not have. We can't go and register somewhere else for PrEP, because [program staff] don't write the next [refill] date. If they did, we would go.*" (6007, 36 years, discontinued)

Additionally, there was a reluctance among women to proactively contact program staff, due to a misunderstanding that the program had a systematic way of communicating upcoming refills and delivery.

> "*I thought of calling, but I thought you follow 'stage by stage. I used to think about calling, but I just decided to wait.*" (4005, 39 years, discontinued)

**Theme: Burdens of opportunity costs, financial strain, and side effects.** Several women responded they were 'too busy' to call the program and follow up on refills, which pointed to an opportunity cost of traveling to a drop-in center and waiting to receive medication, as well as the actual costs of transport. They felt that coming to the program's drop-in centers to pick-up the pills was a burden. This was especially true for women who experienced strong discrimination from government health facilities, who were still building up their trust with the KPIF program.

Additionally, side effects such as increased appetite and weight gain were frequently cited as reasons for discontinuing oral PrEP. These physiological changes posed significant challenges, particularly in the context of poverty, as they exacerbated the financial strain by increasing food consumption needs. This economic burden compelled some women to discontinue PrEP and seek alternative ways to protect themselves from HIV, such as condoms freely provided by the program. One woman highlighted this shift in her strategy for protection due to the side effects and her financial situation:

> "*[PrEP] gave me appetite too much. Sometimes I didn't have money to buy food so I stopped. But now, whether he [client] wants or he doesn't, I always use protection.*" (6009, 22 years, discontinued).

Although this participant had the best of intentions to use condoms, invariably sex workers felt forced to engage in condomless sex. This was either because of fear of violence from clients, or financial stressors that motivated them to engage in condomless sex which pays higher than sex with condoms.

One peer-navigator described how she was able to decline client requests for condomless sex because of her financial stability from multiple income streams, including employment in the program,

> "*it's because at least am a bit financially independent … so even if someone comes, they tell me, "No, let's have sex [and] I'll give you so much, let's not use protection"... it's a 'No' for me.*" (4001, 23 years, peer-navigator)

These experiences highlighted the need for tailored support mechanisms within PrEP programs to address and manage side effects, and to ensure that financial or physiological barriers would not impede continued PrEP use while women still had the desire to take the drug.

**Motivation. Theme: Observability–Personal experiences of PrEP effectiveness.** Women who continued to persist on PrEP perceived the effectiveness and benefits of PrEP in their life, especially in comparison to their peers– illustrating its impact on their lives, and the lives of others within their community. The following quote from a woman who continued PrEP illustrated the concept of "observability" in the diffusion of innovations theory, where the benefits of a new technology (like PrEP) were visible and influential in encouraging its adoption among others [29].

*"You will find that I introduce someone to prostitution, but because of ignorance [not being as careful as me]; they will start taking [ARVs], but me I will be okay and continue to be fit. She leaves me taking PrEP, then...she starts taking ARVs." (6004, 31 years, continued)*

This comparison underscored the participant's belief in PrEP as a preventive measure that had kept them "fit" and free from the need for lifelong ARVs, despite their high-risk lifestyle. The participant's use of the term "ignorance" highlighted a perceived lack of awareness or misunderstanding about the benefits of PrEP among some peers, which they felt had led to poorer health outcomes for others.

As in the previous quote, there was a psychological comfort and security derived from taking PrEP. Women expressed a sense of reassurance and reduced anxiety about acquiring HIV:

*"I use PrEP because then I know everything is fine with me." (6005, 26 years, continuer)*

PrEP was seen, not only a medical tool, but also as a means of psychological assurance for those at high risk of HIV. By showcasing tangible health benefits and offering peace of mind, PrEP effectively demonstrated its value, promoting its wider acceptance and use among sex workers.

**Theme: Oral PrEP offers personal empowerment and health protection in work environments of known HIV acquisition risk.** Oral PrEP was seen by many women as a tool for personal empowerment and health protection in their high-risk work environment. The most salient enablers to initiating and staying on PrEP included internal motivations and perceived personal benefits, such as the desire for personal safety, the acknowledgment of the 'high risk' nature of their job, and the value placed on maintaining a negative HIV status as a form of self-care and empowerment.

For many women, the decision to continue taking PrEP was directly linked to their experiences and the realities of their work. One participant explained,

*"I continued to take PrEP because…sometimes, even if you are not drunk, some clients want 'live' (condomless) sex, so if you don't take PrEP, there is nothing [to protect you]." (4006, 31 years, continuer)*

This highlights the immediate risk factors and the protective role PrEP plays in situations where condom use was not always possible.

Understanding the broader health implications, some women also recognized that acquiring other sexually transmitted infections (STIs) increased their risk of HIV. One peer-navigator shared her rationale for starting PrEP:

*"I also used to have a lot of STIs, so I knew that I was also at risk of getting HIV." (6002, 35 years, peer-navigator)*

Furthermore, the uncertainty surrounding the HIV status of their clients added another layer of risk, which PrEP helps mitigate. As one peer-navigator described,

*"I heard that PrEP helps in protecting against HIV, we are in a chain where we don't know everyone's status, so I started taking it to protect myself." (4002, 26 years, peer-navigator)*

This sentiment reflected the widespread concern about unknowingly being exposed to HIV, and the empowerment women experienced from taking proactive measures.

In summary, the women's reflections and decisions to use PrEP illustrated a clear understanding of their risk environment, and a strong drive towards personal health and empowerment. By taking PrEP, they not only protected themselves from HIV, but also exercised control over their health and well-being in a high-risk occupation.

**Theme: Compatible and flexible with lifestyle.** The compatibility and flexibility of oral PrEP with the lifestyles of sex workers emerged as strong motivators for its initiation and continued use. One key aspect is the ability to start and stop PrEP based on personal circumstances and perceived risk, which allowed them greater control over their health management.

The flexibility of PrEP allowed women to adapt its use based on changes in their personal lives, such as finding a stable partner. As one participant noted about her decision to start PrEP, and her ongoing motivation to continue it:

*"I found out that you can stop PrEP when you find a man that wants to marry you." (4008, 37 years, continuer)*

This highlighted that for some women, the option to discontinue PrEP upon entering a committed relationship felt reassuring and practical. However, viewing PrEP as protection that is only necessary for sex work may create a false sense of security, as HIV risk also exists in stable or marital relationships.

Moreover, the adaptability of PrEP to their fluctuating HIV infection risk was appealing. One participant shared,

*"I usually stop and start [PrEP] whenever I see that I have stopped running around too much." (4001, 23 years, peer-navigator)*

Here, she implies that she adjusted her PrEP usage based on the number of clients, and her perceived risk of HIV acquisition. This adaptability not only supported initial uptake, but also motivated sustained use as circumstances changed.

**Determinants of LAI-PrEP**

**Capability. Theme: Misconceptions and knowledge gaps among peer-navigators and peers.** Even before LAI-PrEP rolled out, misconceptions already existed among peer-navigators regarding the relationship between missing LAI-PrEP doses, seroconversion (the period during which HIV antibodies develop and become detectable), and the subsequent effectiveness of ART.

*"We went to a workshop somewhere. They told us once you start, then you miss a jab, then you seroconvert from PrEP to ART. Meaning ART will never work in your body. That's what we were taught. So that's the disadvantage of injectable PrEP." (6003, 29 years, peer-navigator)*

Women already anticipated misconceptions among their peers, and were concerned that LAI-PrEP might conceal the virus, making it undetectable for those already infected with HIV.

*"There may be a lot of misconceptions among people. People may think PrEP hides the [HIV] virus." (6001, 23 years, peer-navigator)*

Addressing these misconceptions was important, since women recognized their own knowledge gaps on how LAI-PrEP worked, and would rely on their peers to build trust and understanding.

*"I wouldn't take the injectable PrEP, unless my friend told me it was OK. Then I would. I'm afraid because I wouldn't know how it works." (4006, 31 years, continuer)*

**Opportunity. Theme: Inconsistent supply chain availability.** One participant was concerned about the consistent availability of LAI-PrEP. The fear that dependency on a scheduled injection could be disrupted by supply chain issues, lead to anxiety about having to revert to oral pills, which may be seen as a less desirable option. This concern reflected broader logistical challenges in healthcare systems—rooted in past stock disruptions of oral PrEP and family planning injectables—while also highlighting the psychological and practical adjustments required when switching between PrEP formulations.

*"The injectable PrEP may not be available when you need it... When it's your date to take it again … Or it can run out when we are already used to it and that can mean getting back to pills." (4002, 26 years, peer-navigator)*

Inconsistency in the supply of LAI-PrEP could undermine their trust in the healthcare system, and disrupt women's work routines. This inconsistency had the potential to cause significant stress, as the anticipation of potential shortages created uncertainty around maintaining effective HIV prevention. Transitioning back to oral pills may be seen as a step backward, both in terms of convenience and perceived efficacy. The process of adjusting back to daily oral medication may be daunting for sex workers, especially for those who have become accustomed to the less frequent dosing schedule of LAI-PrEP.

Suddenly stopping LAI-PrEP could create a window period where WESW are at risk of HIV infection, and this could foster the development of resistant strains due to suboptimal drug levels when a woman is infected with HIV. During this time, sex workers who discontinue LAI-PrEP may need to switch back to oral pills to maintain continuous protection, which highlights the importance of consistent PrEP use and proper transition planning to prevent gaps in HIV prevention.

**Theme: Sex worker social support system.** Sex workers operated within complex social networks characterized by both competition and camaraderie. The competitive nature of their environment, driven by the need to secure clients, inherently fostered instability within these networks. The following quote illustrated how WESW would use the community stigma around PrEP to their advantage, in order to entice clients away from sex workers who were using PrEP.

*"...Friends …the ones I stay with…they used to say bad things that I am taking ARVs...sometimes even when you are in a bar, they would discourage clients from coming to me and tell them, "You are going to the one taking ARVs, and if you want, you can come home, you will find the bottle." But I was not discouraged, and I didn't stop taking PrEP." (6009, 22 years, discontinuer)*

Despite this competition, the shared experiences and challenges unique to sex work created a basis for mutual understanding and support among workers. This solidarity was crucial for navigating the risks associated with their profession, including personal safety and protection from violence and rape. Moreover, within the context of HIV prevention, this solidarity manifested as a practical support system for PrEP adherence. The practice of sharing PrEP pills among peers who may forget or lack access to their medication underscored the role of social support as a key facilitator in maintaining PrEP regimes. However, the transition to LAI-PrEP may potentially challenge this form of solidarity. Unlike pill sharing, injectable PrEP cannot be shared among peers, potentially leading to a loss of a critical support mechanism within these networks.

*"Sometimes we share tablets, but if in jab form, then no sharing." (6001, 33 years, peer-navigator)*

**Motivation. Theme: Alleviates daily pill burden.** WESW showed a strong preference for LAI-PrEP, facilitated by KPIF-supported peer networks and service delivery platforms. There was a consistent sense that injectable PrEP

would be seamlessly integrated into their lifestyle without the constant reminder or the need for daily action, particularly beneficial for those who were missing doses due to alcohol use.

*"I wouldn't be forgetting." (4009, 23 years, discontinued)*

One peer-navigator described women reporting pill aversion, and the mention of the oral PrEP pill's size as a deterrent suggested that the physical act of taking pills was a barrier to persistence for some, independent of any associated stigma or lifestyle factors. The preference for an injectable option pointed to a potential for increased acceptance and persistence on PrEP, particularly among those who are generally resistant to taking pills.

*"Some of us don't like taking pills even when we are sick because of the size of the pills, but with injectable it will be good." (6003, 29 years, peer-navigator)*

**Theme: May or may not be compatible with occupation and fit into lifestyle**. Injectable PrEP acceptance involved weighing the efficacy of the intervention against the side effects, especially those that interfere with their ability to engage in sex work. Most of the concerns about LAI-PrEP included potential side effects: weight gain, which affected their ability to attract clients, and prolonged menstrual periods which reduce the number of days they can engage in vaginal sex. Other side effects of concern were vomiting, diarrhea, nausea, and injection site pain.

There were further challenges and concerns associated with the logistical aspects of using long-acting injectable (LAI) PrEP, particularly concerning the lifestyle and work demands faced by sex workers. There was inherent uncertainty and mobility in the lives of sex workers, who may have needed to travel frequently and unexpectedly for work. Women were anxious about missing appointments due to their travel, and it's clear that their preference was to receive services through the KPIF program. There was an implied reluctance to get the injectable from other government clinics in the areas where a woman was traveling to.

*"The disadvantage is that maybe that day you have gone on a journey, and your appointment date has come but you're out, what can you do? At least you can carry the pill and go with it." (4005, 39 years, discontinued)*

This scenario contrasted with the flexibility offered by oral PrEP, which could be taken along on travels, ensuring the continuity of protection against HIV without being tied to a specific location or healthcare provider.

**Theme: PrEP stigma.** The confusion between PrEP drugs and ARVs contributed significantly to PrEP stigma, intertwining it with HIV stigma. PrEP medication contained some of the same active ingredients found in ARVs, leading to misconceptions that individuals on PrEP were HIV-positive, or that PrEP was used exclusively by those with high-risk behaviors–thus stigmatizing its use. This misperception could exacerbate the challenges individuals face when deciding to use PrEP, as they may fear being mistakenly perceived as having HIV. The overlap in medication ingredients thus fueled both PrEP and HIV-related stigmas, impacting individuals' willingness to seek and adhere to PrEP as a preventive strategy. Two women described this phenomenon:

*"When you get a jab, no one can see you or know that you have taken PrEP. With the pills, they can compare bottles and think that you are taking ARVs." (6008, 27 years, discontinued)*

*"For some men, when they see you take PrEP, they become suspicious and curious, but when you jab, they can't even know you have taken PrEP." (4005, 39 years, discontinued)*

Women felt empowered by the anonymity when their preventative healthcare choices remain private, safeguarding them from societal judgment and stigma. Many preferred LAI over oral PrEP because *"nobody can know that you are on PrEP"*

---

*(6006, age: 21 years, continuer)*, and because it potentially reduced anticipated stigma, as well as the ease of regimen persistence and adherence.

WESW described a sense of enhanced safety and well-being with injectables that might not be there with oral PrEP. This highlighted the dual aspect of physical health protection from HIV and the reduction of risk from external threats, such as violence, by minimizing visible signs of PrEP usage. Women discussed the anxiety of missing doses when *"a client gets you and keeps you" (4007, 31 years, continuer)* for several days, and fear of clients *"even beating you" (4006, 31 years, continuer)* if they found pills that could be mistaken for antiretrovirals (ARVs).

## Discussion

Our study found that education by peer-navigators and program staff is crucial when providing trustworthy HIV services, and promoting PrEP uptake among sex workers who have faced widespread discrimination in government clinics. Our study also highlights the significant impact of persistent knowledge gaps and misconceptions about PrEP adherence and alcohol use as barriers among sex workers in Zambia. We also found that both trust in program staff, and reliable service provision, facilitated continued use, with a preference for PrEP delivery to them. Social support systems were mixed, providing both aid and competition. Not only that, but we further identified that personal empowerment and health protection motivated PrEP use, with LAI-PrEP preferred for both reducing daily pill burden, and the stigma of oral PrEP as HIV medication. However, concerns about inconsistent supply, and misconceptions about LAI-PrEP, present potential barriers to uptake.

Our findings on the barriers to oral and injectable PrEP initiation and persistence among WESW in Lusaka echo similar challenges documented in other African contexts, including stigma, challenges accessing healthcare, and misconceptions about PrEP [23,30]. The expressed preference for LAI-PrEP over oral formulations aligns with a growing demand for more discreet and convenient HIV prevention methods [23]. This preference underscores a potential shift in prevention strategies that could significantly reduce new HIV infections among sex workers and their sexual partners in Southern Africa.

In the following sections we outline potential PrEP interventions for the Zambian Key Populations Investment Fund informed by the COM-B Model [S1 Text].

### Addressing misconceptions and education gaps

Increasing uptake and engagement with both oral PrEP and LAI-PrEP requires targeted education for all program staff, including healthcare providers and peer navigators. This education should clearly differentiate the mechanisms and actions of PrEP and ART, and dispel myths about PrEP.

Providing program staff with talking points or brochures on the common misconceptions during HIV testing and counseling can enhance understanding. Techniques like role-playing and Q&A sessions can educate about PrEP's effectiveness, even when taken with alcohol or during pregnancy [31]. Care should be taken to balance messaging around PrEP's flexibility, with education on HIV risk across different relationship types (client versus committed monogamous partnerships) to ensure informed decision-making. These initiatives are crucial for correcting misinformation, alleviating fears about discontinuing PrEP, and maintaining trust in ART, particularly as PrEP initiation may be the first interaction WESW have with the program.

As LAI-PrEP becomes more widely available, peer-led education programs can further clarify its role, address its side effects, and emphasize the need for adherence to dosing schedules. Peer navigators and "PrEP champions" who have benefited from oral PrEP could lead interactive workshops tailored to different literacy levels and languages. Facilitating peer-led group discussions allows for sharing experiences and learning in a supportive environment [32,33].

### Enhancing PrEP delivery systems

Considering the preference for LAI-PrEP, it is critical for policymakers and healthcare providers to develop effective rollout strategies. These should ensure an adequate supply of injectables, proper training for healthcare workers, and protocols

that accommodate the specific needs of WESW. Addressing fears of inconsistent supply and ensuring clear communication about alternative options can mitigate concerns [30]. Providing standardized refill cards and using phone calls and SMS reminders to communicate PrEP refill dates can enhance engagement. Additional counseling on the importance of persistence during one-month follow-ups post-PrEP initiation has also been effective in programs tailored for sex workers in South Africa [34,35].

Younger women who sell sex often face greater adherence challenges, necessitating support tailored to their age group and lifestyle [36]. PrEP delivery programs for adolescent girls and young women should offer a choice of adherence support methods, including counselor- and peer-based options, as well as in-person and mHealth options, to suit individual preferences and circumstances [37].

## Make PrEP delivery more convenient

Confidentiality, privacy, and worthiness are key qualities sex workers seek in PrEP services [38]. The program can cultivate this trust by making PrEP service delivery more convenient. This includes providing PrEP and HIV testing services at times and locations that do not conflict with WESW work schedules, especially since travel is a lifestyle factor that is known to interfere with proper medication intake among sex workers [35]. Venue-based outreach offering injectable PrEP at locations like KI homes can align with their needs. Integrating financial support mechanisms like transport vouchers can reduce the opportunity costs of accessing PrEP services, shown to create demand for family planning and HIV self-testing. Offering flexible service hours, multiple discreet locations, and adaptable refill schedules can further support PrEP adherence. Developing these strategies with input from WESW and their advocacy groups ensures interventions are not only contextually relevant, but also have the buy-in necessary for effectiveness.

## Integrating socioeconomic considerations into health programs

The side effects of PrEP, combined with the opportunity costs and logistical barriers to accessing refills, illustrate the need for holistic health programs. Women often engage in sex work due to poverty and insecurity, making HIV prevention a lower priority than basic needs [7]. Holistic health programs should address both medical and economic challenges to ensure sustainable intervention strategies, particularly for LAI-PrEP. Failure to adhere to LAI-PrEP schedules can risk HIV drug resistance. Practical solutions include holistic health education beyond HIV prevention, teaching financial literacy and nutrition, and offering microfinance loans or grants to empower women economically which provides them with alternative income sources.

## Navigating alcohol use and persistence challenges

Alcohol use significantly impacts PrEP adherence, as indicated by studies in Kenya where substance use disorders hinder timely medication intake [39]. Educational programs should stress the importance of maintaining medication effectiveness, and should include strategies like peer support, buddy systems, and behavioral regulation tactics linking medication intake with daily activities that are less likely to be disrupted by alcohol (i.e. wearing a wig or putting on clothes) [40]. A person-centered approach to healthcare would provide substance use and mental health counseling alongside HIV prevention services.

## Societal influence on health decisions

The interplay between societal stigma and personal health decision-making highlights the unique challenges faced by WESW in Zambia. Despite the competitive nature of their social networks, support from these networks is crucial for enhancing personal safety and facilitating PrEP persistence. PrEP stigma is driven by ongoing stigma around HIV, limited knowledge about PrEP and HIV, and ongoing stigma around adolescence, being a single woman, and sexuality itself [41].

Social support encourages open discussion of stigma and discrimination, reducing distress. Engaging influential community members as opinion leaders can promote positive attitudes towards PrEP, and this may include brothel leadership who have played facilitatory roles in condom use among WESW and their clients [42]. Successful digital stigma reduction interventions among Black women in the US and Brazil show that trusted individuals can foster confidence in PrEP [43,44].

## Limitations

This study offers significant insights, but has several limitations that affect the breadth of its applicability. Primarily, our research focused on sex workers in Lusaka, Zambia who solicit clients from streets, brothels, bars, and nightclubs. This does not encompass the experiences of 'high-class' sex workers who operate from more discreet venues such as massage parlors, hotels, professional offices, or university settings. Consequently, our findings may not be generalizable to all sex worker demographics within Zambia or other regions.

The relevance and importance of barriers and facilitators are context-dependent, and no barrier or facilitator is universally applicable [45]. As injectable PrEP becomes more accessible, further qualitative research is essential. Collaboration with community advisory boards can help assess the adoption, adherence, and effectiveness of injectable PrEP among different sub-groups of women in sex work. Such studies will clarify the full potential of injectable PrEP in HIV prevention and address gaps identified in this initial research.

We acknowledge that rapid analysis differs from analyzing full transcripts, because it involves a more focused and expedited coding process. However, studies suggest that while rapid analysis may not capture all nuances, it reliably identifies key themes [24,25]. A single researcher conducted all interviews due to resource constraints, however, their extensive experience in HIV research, and engagement with the sex work community with the guidance of the community advisory board and key community members, as well as the use of reflexivity, helped mitigate potential interviewer bias.

## Conclusion

This study reveals the multifaceted dynamics influencing PrEP engagement among WESW in Lusaka. While individual motivations and programmatic support significantly promote uptake, substantial socio-economic and logistical barriers hinder sustained use. The preference for LAI-PrEP, due to its discretion and elimination of pill burden, aligns with a broader trend towards more manageable HIV prevention strategies. However, WESW are concerned about the new complexities with LAI-PrEP, including scheduling constraints and potential supply interruptions, which must be navigated carefully to maximize its effectiveness.

Future interventions should focus on expanding access to PrEP through targeted, context-specific strategies that address both the medical and social needs of sex workers. This includes improving education around PrEP to prevent misinformation, particularly concerning its distinction from ART, and enhancing service delivery to accommodate the unpredictable lifestyles of sex workers. As PrEP options evolve, continuous research is required to assess their impact and acceptance, ensuring that advances in HIV prevention translate into tangible benefits for those at the highest risk.

Adopting a person-centered care approach is crucial, and it aligns with calls for transformative global health approaches that challenge existing power dynamics and engage marginalized populations in designing solutions for their own communities[46,47]. This includes integrating holistic strategies that address both health and socio-economic challenges, such as providing comprehensive support services, financial assistance, and flexible scheduling which is tailored to the unique needs of WESW. Such holistic approaches are vital for the broader integration of PrEP into HIV prevention programs in Zambia and similar settings, potentially setting a precedent for global health practices.

This research not only sheds light on a marginalized community but also urges policymakers, healthcare professionals, and social support systems to acknowledge and address the unique challenges faced by sex workers in Lusaka. It is a call to action for a more inclusive and empathetic approach towards a group too often overlooked and underserved.

## Supporting information

**S1 Checklist.  Consolidated criteria for reporting qualitative studies (COREQ): 32-item checklist.** (DOCX)

**S1 Data.  Memo Data.** (XLSX)

**S1 Table.  Potential PrEP Interventions for the Zambian Key Populations Investment Fund Informed by the COM-B Model.** (DOCX)

**S1 Text.  In-Depth Interview Guide for Women Engaged in Sex Work or Peer-navigators.** (DOCX)

**S2 Checklist.  Inclusivity in Global Research.** (DOCX)

## Acknowledgments

The authors would like to acknowledge the infrastructure support provided by the Centre for Infectious Disease Research in Zambia (CIDRZ) and the Key Populations Investment Fund (KPIF) program. The authors would also like to thank peer-navigators and key leaders in the sex-work community for their assistance in recruiting study participants. Additionally, we thank Mr Kumar Srinivasan and Ms Ritika Kumar for their transcription and analysis support.

## Author contributions

**Conceptualization:** Ramya Kumar, Wendy Barrington, Ruth Zyambo, Martin Zimba, Kenneth Mugwanya, Michael Herce, Maurice Musheke, Deepa Rao, Anjali Sharma.

**Data curation:** Ramya Kumar, Chisomo Mwale, Patricia Maritim, Jamia Phiri.

**Formal analysis:** Ramya Kumar, Chisomo Mwale, Patricia Maritim, Jamia Phiri, Deepa Rao.

**Funding acquisition:** Ramya Kumar, Kenneth Mugwanya, Michael Herce, Maurice Musheke.

**Investigation:** Ramya Kumar, Jamia Phiri, Ruth Zyambo, Kenneth Mugwanya, Maurice Musheke, Anjali Sharma.

**Methodology:** Ramya Kumar, Patricia Maritim, Jamia Phiri, Wendy Barrington, Ruth Zyambo, Martin Zimba, Michael Herce, Anjali Sharma.

**Project administration:** Ramya Kumar, Martin Zimba, Maurice Musheke, Anjali Sharma.

**Resources:** Ramya Kumar, Martin Zimba, Michael Herce, Maurice Musheke, Anjali Sharma.

**Software:** Ramya Kumar, Anjali Sharma.

**Supervision:** Ramya Kumar, Martin Zimba, Kenneth Mugwanya, Michael Herce, Maurice Musheke, Deepa Rao, Anjali Sharma.

**Validation:** Ramya Kumar, Chisomo Mwale, Patricia Maritim, Anjali Sharma.

**Visualization:** Ramya Kumar, Patricia Maritim.

**Writing – original draft:** Ramya Kumar, Chisomo Mwale, Patricia Maritim, Jamia Phiri, Kenneth Mugwanya, Deepa Rao, Anjali Sharma.

**Writing – review & editing:** Ramya Kumar, Chisomo Mwale, Patricia Maritim, Jamia Phiri, Wendy Barrington, Ruth Zyambo, Martin Zimba, Kenneth Mugwanya, Michael Herce, Maurice Musheke, Deepa Rao, Anjali Sharma.

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
