## [Decision Letter · Decision Letter 0]

PGPH-D-24-02335

Beyond the Pill: Understanding Barriers and Enablers to Oral and Long-Acting Injectable PrEP Among Women in Sex Work in Zambia.

Dear Dr. Kumar,

Thank you for submitting your manuscript to PLOS Global Public Health. After careful consideration, we feel that it has merit but does not fully meet PLOS Global Public Health’s publication criteria as it currently stands. Therefore, we invite you to submit a revised version of the manuscript that addresses the points raised during the review process.

The manuscript received favorable feedback from the peer review process. The evaluations highlighted the study's comprehensive analysis of perceptions of Long-Acting injectable PrEP among sex workers in sub-Saharan Africa. Although some findings reinforced existing knowledge, the research makes a distinct contribution by revealing ongoing challenges in PrEP implementation and demonstrating how injectable options could reduce stigma through discreet administration. The reviewers appreciated the community-engaged approach to developing research instruments and suggested major revisions to enhance clarity. Overall, the work was deemed a timely and valuable addition to the field, presenting complex insights in an accessible format.

As editor, I consider that the manuscript has the potential to be published but may not be accepted if you do not address substantive issues.

The essential revisions are as follows:

Methods: Both reviewers have provided suggestions to clarify the methods section, with Reviewer 1 requesting clarification about the COM-B framework usage and analysis process, while Reviewer 2 raises concerns about potential bias from using a single interviewer. Both appreciate methodological transparency, with Reviewer 1 specifically commending the reflexivity statement.

Results: A key overlapping concern is about interpretation and evidence: reviewers have noted instances where interpretations lack direct support from participant quotes. Reviewer 1 questions specific interpretations around trust-building and PrEP discontinuation, while Reviewer 2 suggests condensing the dense results (15 themes) into a more coherent narrative. Reviewers have highlighted the need to better ground interpretations in participant data, and authors have to better distinguish between participant experiences and researcher observations.

Discussion: Reviewers have addressed literature engagement and limitations, with Reviewer 1 requesting more references for intervention recommendations and Reviewer 2 challenging claims of novelty regarding alcohol use findings by citing existing literature. You should include references to the relevant studies mentioned in your new version. They also agree that certain interpretive content currently in the Results section would be better placed in the Discussion.

I also recommend that the authors respond to any minor comments made by the two reviewers.

We look forward to receiving your revised manuscript.

Kind regards,

Marie Meudec, PhD

Academic Editor

Journal Requirements:

2. Please ensure that the Title in your manuscript file and the Title provided in your online submission form are the same.

3. We have amended your Competing Interest statement to comply with journal style. We kindly ask that you double check the statement and let us know if anything is incorrect. 

4. In the online submission form, you indicated that "Due to the sensitive nature of the research involving a small sample of marginalized women and the need to protect participant confidentiality, the interview data are not publicly available. Access to de-identified transcripts or other supporting information may be provided upon reasonable request, subject to ethical approvals and institutional review board guidelines". 

a. In a public repository, 

b. Within the manuscript itself, or 

c. Uploaded as supplementary information.

Reviewers' comments:

Reviewer's Responses to Questions

**Comments to the Author**

1. Does this manuscript meet PLOS Global Public Health’s publication criteria? Is the manuscript technically sound, and do the data support the conclusions? The manuscript must describe methodologically and ethically rigorous research with conclusions that are appropriately drawn based on the data presented.

Reviewer #1: Yes

Reviewer #2: Yes

2. Has the statistical analysis been performed appropriately and rigorously?

Reviewer #1: N/A

Reviewer #2: N/A

3. Have the authors made all data underlying the findings in their manuscript fully available (please refer to the Data Availability Statement at the start of the manuscript PDF file)?

Reviewer #1: Yes

Reviewer #2: Yes

4. Is the manuscript presented in an intelligible fashion and written in standard English?

Reviewer #1: Yes

Reviewer #2: Yes

5. Review Comments to the Author

Reviewer #1: General:

Thank you for allowing me to review this interesting and well-written article on a very relevant, timely and important subject. I commend the authors for their thoughtful approach on conducting the research, which generated very insightful and nuanced insights, conveyed in a very structured and clear way. My comments are mainly minor, aiming to clarify and further improve the manuscript.

Abstract:

1. Line 40 (conclusion): the abbreviation “KP” is used but has not been introduced earlier. I suggest to spell this out.

Introduction:

2. The authors could consider giving some absolute numbers on women who sell sex affected by HIV on a global (and/or local) scale, to give an indication of the scope of the problem.

3. Line 74: spelling mistake “three times higher at 33.3%”.

4. Line 76: do you have a reference to support the statement that “failure to protect WESW will prolong the pandemic indefinitely at a huge cost” (e.g. is this based on modelling or personal projections)?

5. Line 78: PrEP is not only recommended by WHO for key populations but all at increased risk of HIV.

6. Line 83-92: I felt this paragraph could have benefited from some more interaction with the existing literature on (structural) barriers to PrEP. It would be good to include some references here.

7. Line 93-96: the authors may want to consider giving some more detail on the geographic regions and populations among which the LAI-PrEP trials were conducted.

8. Line 117: It was not clear to me what was meant by “reachable moments” – consider just talking about “reach”?

Methods:

9. Line 129: I was wondering whether the aim of the parental study could be mentioned. Reason being that I was interested to know whether there was quantitative data available on the uptake and continuation rates of PrEP among WESW in this context. This could be useful to contextualise some of the qualitative findings as well.

10. Consider including a visual for the COM-B model?

11. Line 153: I would suggest to use “effectiveness” instead of “efficacy” when talking about real-world implementation of these interventions.

12. Participant selection: were participants sampled from the quantitative survey, or how was this done?

13. Data analysis: It was not entirely clear to me whether the final analysis was based on the summary memos (step 1) or on the generated verbatim transcripts? In addition, am I understanding it correctly that the codebook included the COM-B constructs as higher-level (deductive) categories, but that themes under these categories were still inductively delineated? If so, I would consider phrasing it like this (and not refer to the COM-B construct as “themes” to avoid confusion).

14. Nice to include a critical reflexivity statement!

Results:

15. Line 319-322: this difference in trust among participants in the program’s services is attributed (by the authors?) to time (“building trust takes time”). However, is this an inference that was made from the data or was this an explanation ascribed by the authors? As a general comment, I would be careful in the Results section to ground statements explicitly in the data, and reserve discussing implications or less obvious potential explanations for the Discussion section.

16. Line 336: “even among initiators”- redundant, as you have to have initiated to be able to discontinue?

17. Line 353-362: I thought this part was a bit more speculative, as the argument (although valid on its own) is entirely based on an uncertain explanation of PrEP discontinuation during pregnancy. Why was it considered likely by the authors that PrEP discontinuation was attributed to pregnancy here? I would specify or if insufficient data to back this, leave it out.

18. Line 431-432: there seems to be an editing issue with the sentence here.

19. Line 523: I thought this was a bit of a difficult quote to follow as PrEP are also ARVs. Is this participant talking about HIV-positive colleagues here when talking about ARVs? Consider using a different quote if there are any that can make this point clearer?

20. Line 575: As I was reading about this theme on flexibility, I had the reflection that considering PrEP only as an intervention relevant to the work setting (e.g. to be used when engaging in sex work activities) and not so much to sexual relationships with (stable) private partners might also carry a risk of generating a false feeling of safety, as HIV can also be acquired outside more “commercial” relationships. The authors may want to comment on this in the manuscript.

21. Line 600: is there a word missing (e.g. “even before LAI-PrEP rollout”)?

22. Line 622: do the authors have any insights as to why consistent availability of LAI-PrEP was considered to be more of an issue or concern as compared to the consistent availability or oral PrEP? Was this rooted in experiences of stock disruptions for other injectable products (e.g. for family planning)?

23. Line 641: rather than infection by resistant strains, I think that suddenly stopping LAI-PrEP could foster the development of resistant strains due to suboptimal drug levels when infected with HIV.

Discussion:

24. Generally well-written and balanced discussion. I would suggest including some more references. For instance, Line 770 (evidence for role-playing techniques or Q&A sessions?), Line 778-779 (evidence for peer-led group discussions?), and Line 825-829 (evidence for related interventions?).

25. Limitations: any limitations to the rapid analysis approach (cfr. what would a fully fletched thematic analysis have added to the analysis)? Do the authors consider any limitations regarding the self-reported nature of the data (e.g. related to social desirability or any other bias), or was this risk considered to be low in this context?

Reviewer #2: This manuscript describes a qualitative study of PrEP enablers and barriers in the setting of injectable cabotegravir rollout, including for female sex workers, in Zambia. Whereas the findings have largely been reported in previous research, the paper is a valuable addition to the literature, given the lack of literature on sex worker perceptions of long-acting cabotegravir in sub-Saharan Africa. The paper also serves as a helpful reminder that the same barriers encountered during oral PrEP rollout persist as long-acting PrEP is scaled up. Of interest is the perception that injectable PrEP is empowering because it can be used discreetly and circumvent the HIV-associated stigma associated with oral PrEP. A strength of the study is that community based participatory research methods were used to design and pilot-test the interview guide.

My overarching comment is that the “arguments” in the theme descriptions should primarily reflect the viewpoints, experiences, and voices of the interviewees (participants’ voices) rather than the researcher’s observations (analyst's voice, e.g., lines 353-362, 640-644), which are more appropriately placed in the discussion section. Additionally, the results are pretty dense (15 themes spanning 14 pages), and it would be helpful if the authors distilled their key findings into a qualitative results summary that tells a coherent story for the reader's benefit. I have a few comments which the authors might find helpful.

1) Interpretation: The description of themes should be directly supported by evidence from participant quotes that illustrate the lived experiences behind the theme. However, thematic descriptions are only sometimes backed up by evidence in illustrative quotes. For example, the elaboration in lines 690-696 is not supported by evidence from quotes and appears to reflect the researchers’ observations. Medical mistrust (lines 400, 415) is not evidenced by the quotes (lines 410-413, 420-423), which describe enacted interpersonal stigma in healthcare settings.

2) Methods – Interviewer effect: A limitation of the study is that a single interviewer conducted all interviews, and there is potential for bias because one person's perspective could have influenced the questions asked, potentially skewing the results. This should be acknowledged in the Limitations.

3) The Results section is written in present and past tense. Using the past tense is customary since you describe findings derived from completed data collection and analysis.

4) Results, Theme 3 (alcohol use is part of the lifestyle, but incompatible with daily pill adherence). The quotes provided support alcohol use as a coping strategy but do not provide direct evidence that alcohol use is incompatible with adherence. The third quote alludes to forgetting pill taking when drunk, but this is not the same as “incompatibility with daily pill adherence.”

5) Results, lines 353-362: The authors provide an anecdote about a pregnant participant going into labour shortly after completing her in-depth interview. Since these observations about person-centered care are not directly derived from participant interviews, they are better placed in the Discussion.

6) Discussion: The authors state that their study is the first to describe significant knowledge gaps and misconceptions about PrEP adherence and alcohol use as barriers. At least 18 studies on PrEP awareness and interest among people with unhealthy alcohol use were reviewed in a scoping review (AIDS Behav. 2020 Nov 20;25(6):1777–1789), including among young women in Kenya (AIDS Res Hum Retroviruses. 2016;32((Supplement 1)):384) and sex workers in Zambia (BMJ Public Health 2024;2:e000483).

7) Line 176: “Women are sometimes given the date for next PrEP refill…” Using the word “sometimes” makes it seem that appointment dates are optional.

8) Lines 202-203: “will be available in Zambia next year”. For specificity, perhaps replace “next year” with 2024.

9) Lines 231-232: Incomplete sentence: “health education, particularly in encouraging the acceptance and use of PrEP among WESW.”

10) Lines 583-584: The word “when” at the beginning of the sentence changes its meaning. Removing it implies the interviewee would stop taking PrEP after she found a suitor, which is probably the intended meaning.

11) Line 600: Add a comma after LAI-PrEP to improve sentence comprehension.

12) Lines 640-644: The analyst’s voice should be in the Discussion section.

13) Line 722 should read, “Two women described this phenomenon.”

14) Line 755: “Our findings on the barriers to oral PrEP initiation…” What about injectable PrEP, which was a key focus of the study?

15) Lines 755-757 could be better referenced with other studies of PrEP misconceptions, e.g., AIDS Behav. 2021 Aug;25(8):2517-2532, Arch Sex Behav. 2021 May;50(4):1729-1742, Front Public Health. 2022 Jun 17;10:691729).

16) Line 842: What does “this may income brothel leadership” mean?

6. PLOS authors have the option to publish the peer review history of their article (what does this mean?). If published, this will include your full peer review and any attached files.

**Do you want your identity to be public for this peer review?** For information about this choice, including consent withdrawal, please see our Privacy Policy.

Reviewer #1: **Yes: **Jef Vanhamel

Reviewer #2: No

---

## [Decision Letter · Decision Letter 1]

Beyond the Pill: Understanding Barriers and Enablers to Oral and Long-Acting Injectable PrEP Among Women in Sex Work in Zambia.

PGPH-D-24-02335R1

Dear Dr Kumar,

We are pleased to inform you that your manuscript 'Beyond the Pill: Understanding Barriers and Enablers to Oral and Long-Acting Injectable PrEP Among Women in Sex Work in Zambia.' has been provisionally accepted for publication in PLOS Global Public Health.

Best regards,

Marie Meudec, PhD

Academic Editor

Thank you for responding to the reviewers' comments. The reviewers and I approve the revisions and therefore accept the new version of your article.

Reviewer Comments (if any, and for reference):

Reviewer's Responses to Questions

**Comments to the Author**

1. If the authors have adequately addressed your comments raised in a previous round of review and you feel that this manuscript is now acceptable for publication, you may indicate that here to bypass the “Comments to the Author” section, enter your conflict of interest statement in the “Confidential to Editor” section, and submit your "Accept" recommendation.

Reviewer #1: All comments have been addressed

Reviewer #2: All comments have been addressed

2. Does this manuscript meet PLOS Global Public Health’s publication criteria? Is the manuscript technically sound, and do the data support the conclusions? The manuscript must describe methodologically and ethically rigorous research with conclusions that are appropriately drawn based on the data presented.

Reviewer #1: Yes

Reviewer #2: Yes

3. Has the statistical analysis been performed appropriately and rigorously?

Reviewer #1: N/A

Reviewer #2: N/A

4. Have the authors made all data underlying the findings in their manuscript fully available (please refer to the Data Availability Statement at the start of the manuscript PDF file)?

Reviewer #1: Yes

Reviewer #2: Yes

5. Is the manuscript presented in an intelligible fashion and written in standard English?

Reviewer #1: Yes

Reviewer #2: Yes

6. Review Comments to the Author

Reviewer #1: I thank the authors for their thoughtful response to my comments, which have been carefully considered. I have no further remarks and once again congratulate the authors on their work.

Reviewer #2: The authors have satisfactorily addressed my comments.

7. PLOS authors have the option to publish the peer review history of their article (what does this mean?). If published, this will include your full peer review and any attached files.

**Do you want your identity to be public for this peer review?** For information about this choice, including consent withdrawal, please see our Privacy Policy.

Reviewer #1: **Yes: **Jef Vanhamel

Reviewer #2: No
